# Community Ageing Research 75+ (CARE75+) REMOTE study: a remote model of recruitment and assessment of the health, well-being and social circumstances of older people

Lesley Brown ©,[1] Anne Heaven,[1] Catherine Quinn,[2] Victoria Goodwin ©,[3] Carolyn Chew-Graham,[4] Farhat Mahmood,[1] Sarah Hallas,[1] Ikhlaq Jacob,[1] Caroline Brundle,[1] Kate Best,[5] Amrit Daffu-O'Reilly,[6] Karen Spilsbury ©,[6] Tracey Anne Young,[7] Rebecca Hawkins,[5] Barbara Hanratty ©,[8] Elizabeth Teale,[5] Andrew Clegg ©[5]

For numbered affiliations see end of article.

**Correspondence to**
Dr Lesley Brown;
lesley.brown@bthft.nhs.uk

## ABSTRACT

**Introduction** The Community Ageing Research 75+ (CARE75+) study is a longitudinal cohort study collecting extensive health and social data, with a focus on frailty, independence and quality of life in older age. CARE75+ was the first international experimental frailty research cohort designed using trial within cohorts (TwiCs) methodology, aligning epidemiological research with clinical trial evaluation of interventions to improve the health and well-being of older people. CARE75+ REMOTE is an extension of CARE75+ using a remote model that does not require face-to-face interactions for data collection in the current circumstances of a global pandemic and will provide an efficient, sustainable data collection model.

**Methods and analysis** Prospective cohort study using TwiCs. One thousand community-dwelling older people (≥75 years) will be recruited from UK general practices by telephone. Exclusions include: nursing home/care home residents; those with an estimated life expectancy of 3 months or less; and people receiving palliative care.

**Data collection** Assessments will be conducted by telephone, web-submission or postal questionnaire: baseline, 6 months, 12 months, 18 months, 24 months, 30 months and 36 months. Measures include activities of daily living, mood, health-related quality of life, comorbidities, medications, frailty, informal care, healthcare and social care service use. Consent will be sought for data linkage and invitations to additional studies (sub-studies).

**Ethics and dissemination** CARE75+ was approved by the National Research Ethics Service (NRES) Committee Yorkshire and the Humber—Bradford Leeds 10 October 2014 (14/YH/1120). CARE75+ REMOTE (amendment 13) was approved on the 18th November 2020. Consent is sought if an individual is willing to participate and has capacity to provide informed consent. Consultee assent is sought if an individual lacks capacity. Results will be disseminated in peer-reviewed scientific journals and conferences. Results will be summarised and disseminated to study participants via newsletters, local engagement events and on a bespoke website.

**Trial registration number** ISRCTN16588124.

## Strengths and limitations of this study

► The Community Ageing Research 75+ (CARE75+) study is a prospective cohort study recruiting older people aged 75 years and over, designed using trial within cohorts (TwiCs) methods.
► Outcome measures were agreed following discussions with academic partners and lay representatives.
► CARE75+ REMOTE will continue to recruit under-served populations including participants from the South Asian community and those with advanced frailty.
► Care home residents are not eligible for the study, aligned with the TwiCs design, meaning that findings cannot be generalised to non-community dwelling older people or those with very advanced frailty.
► CARE75+ REMOTE will provide important data during a time of national pandemic (COVID-19) and will provide a highly efficient, sustainable model of data collection in the future.

## INTRODUCTION

Global demographic projections indicate that there will be two billion people aged over 65 worldwide by 2050.[1] With increasing life expectancy, there are projected to be more older people living with multiple health conditions.[2] Frailty is an especially problematic expression of population ageing, with profound implications for the planning and delivery of health and social care services. Frailty is characterised by age-related decline

BMJ

across multiple physiological systems and vulnerability to disproportionate changes in health after relatively minor health events, such as a minor infection or minor surgery.[3] It is estimated that frailty affects around 12% of people aged≥50 years globally, rising to around one-third of those aged over 80 years.[4] In the last 20 years, there has been a conceptualisation of frailty as an abnormal health state in relation to the ageing process, resulting in the development of robust models and tools to identify and severity grade frailty.[5 6] Increasingly, frailty is considered a long-term condition requiring long-term strategies and interventions.[7] However, the recruitment of older people into research, particularly those with frailty, remains challenging.[8]

The Community Ageing Research 75+ (CARE75+) study was established in 2014.[9] The aim of the study was to establish a longitudinal cohort of older people to investigate ageing, frailty, disability and quality of life in older age. Additionally, the CARE75+study aimed to provide a research recruitment platform for other research studies (sub-studies) to enable the development and evaluation of interventions to improve outcomes for people in later life.

By 2020, 1323 participants had been recruited into the CARE75+cohort. Participants ranged in age from 75 years to 98 years (at baseline assessment) and included people across the spectrum of health and frailty, from varied ethnic and socioeconomic backgrounds. Participants were recruited from a mixture of urban and rural locations across England including Bradford, Leeds, Hull, Scarborough, Newcastle, Durham, Doncaster, Oswestry, Stafford, Wolverhampton, Exeter and Plymouth.

To date, the study has provided a valuable data resource for research on topics pertinent to older people.[10–14] The CARE75+study has also provided a successful recruitment platform for research with older people, with approximately 85% of participants providing consent to be contacted about additional studies (sub-studies). This has enabled stratified sampling and the identification of underserved populations,[15] including for studies on the effect of ageing on skeletal muscle[16]; resourcefulness in later life and the impact of lockdown on the lives of older people during the COVID-19 pandemic.[17]

## CARE75+ REMOTE
In 2019, additional funding was secured for CARE75+via the National Institute for Health Research Applied Research Collaboration, Yorkshire and Humber (NIHR ARCYH)[18] and an updated CARE75+assessment schedule was planned to reflect new research priorities, mainly structured around face-to-face data collection methods, alternating with telephone or web-based data collection at follow-up time points. Outcome measures were agreed following discussions with academic partners and lay representatives through a purposely established Patient and Public Involvement (PPI) Frailty Oversight Group (FOG).[19] The updated CARE75+assessment schedule was designed to provide the necessary data and recruitment

platform for planned studies funded as part of the NIHR ARCYH (https://www.arc-yh.nihr.ac.uk/what-we-do/older-people). However, the advent of the COVID-19 pandemic has necessitated adaption of study procedures to recruit new participants without face-to-face interactions. As a result, the CARE75+REMOTE model has been developed. This model will enable key outcome measures to be captured that do not require face-face interactions. For a description of the different study pathways for the CARE75+study, see table 1.

In addition to supporting ongoing recruitment during the current COVID-19 pandemic, the CARE75+REMOTE model will enhance sustainability of the cohort as a platform for applied epidemiology and novel research involving older people. It is also recognised that research during the COVID-19 pandemic failed to meet the challenges of older people, even though this group was disproportionately affected by the pandemic in terms of restrictions on social contact and mortality.[20] Implementing remote cohort consent, data collection and follow-up methods in advance of a future pandemic will ensure that the ageing research community is better prepared for future research that meets the needs of this underserved population[15] and is cost-effective.

## AIM
The main aim of CARE75+REMOTE is to use remote consent, assessment and follow-up methods to support epidemiological investigation of ageing, frailty, health and well-being and health and social care use in older age. CARE75+REMOTE will also provide a recruitment platform for planned sub-studies within the ARC Older People's programme (https://www.arc-yh.nihr.ac.uk/what-we-do/older-people) and to enable the development and evaluation of future interventions to improve outcomes for older people.

## OBJECTIVES
► To recruit and follow-up older people aged 75+across a range of urban and rural areas using remote methods at 6 monthly intervals for 36 months.
► To support epidemiological analysis relating to healthy ageing and frailty in later life.
► To establish the necessary infrastructure for development and evaluation of interventions to improve outcomes, using TWiCs methods.

## PATIENT AND PUBLIC INVOLVEMENT
The PPI model of engagement, which underpins the CARE75+study, is described in detail elsewhere.[19] For CARE75+REMOTE, we have worked with the FOG to enable researchers to test the CARE75+REMOTE protocol, to ensure it is not overly burdensome and test the postal questionnaire, where multiple iterations were tested prior to an agreed final version.

**Table 1** CARE75+ study pathways

| | CARE75+ | CARE75+ REMOTE | CARE75+2 |
|---|---|---|---|
| Recruitment start | December 2014 | May 2021 | Anticipated start April 2022 |
| Recruitment end | March 2020 | September 2024 | |
| Recruitment target | 2500 across all CARE75+ study pathways by September 2024 | | |
| Recruitment to date | 1323 | 14 | 0 |
| Study status | No new recruitment on this pathway. Follow-ups in progress | On-going with recruitment and follow-ups underway | Not started. REC approval in place |
| Consent | In person, at the participant's home | Telephone (audio-recorded) or postal | In person, at the participant's home. In person, at a general practice location |
| Assessment format | In person, at the participant's home, apart from at 6 months where there is the option of a modified protocol by telephone or web submission. | Postal questionnaire, web submission or by telephone. If there is deterioration in the participant's health, hearing or vision, or cognition, during the course of the study, which makes the above formats problematic, face-to-face assessments will be undertaken if safe and feasible. | In person at baseline, 12 months and 24 months intermediate and final assessments (6, 18, 30 and 36 months) will be conducted by telephone, postal or web submission. Option for in-person assessments to take place at a general practice location. |
| Assessment time points | Baseline, 6, 12, 24, 48 months | Baseline, 6, 12, 18, 24, 30, 36 months | Baseline, 6, 12, 18, 24, 30, 36 months |
| Outcome domains included across all study pathways | Sociodemographic data<br>Frailty<br>Health-related quality of life<br>Activities of daily living<br>Depression<br>Cognition<br>Comorbidities (GP)<br>Medication (GP) | | |
| Additional outcome domains for specific CARE75+pathways | Mobility<br>Muscle strength<br>Pain<br>Hearing<br>Vision<br>Loneliness<br>Resilience<br>Self-efficacy<br>Falls<br>Blood pressure<br>Sedentary behaviour<br>Feeling safe in your neighbourhood | Anxiety<br>Informal care<br>Healthcare and social care use<br>Aids and adaptations | Mobility<br>Physical performance<br>Hearing (satisfaction with hearing aids)<br>Vision<br>Anxiety<br>Informal care<br>Healthcare and social care use<br>Aids and adaptations<br>Loneliness<br>Social isolation<br>Social networks<br>Resilience<br>Self-efficacy<br>Caring responsibilities<br>Carer's quality of life (if applicable)<br>Nutrition<br>Dental health<br>Access to the internet<br>Feeling safe in your neighbourhood |
| Blood samples | Yes (limited sites only) | No | Yes (limited sites only) |
| Notes | Due to the impact of COVID-19, face-to-face follow-up assessments moved to telephone assessments until June 2021 when face-to-face visits were able to resume with specific safety measures in place, and depending on guidance by individual NHS trusts and the research capacity within individual Clinical Research Network organisations. | CARE75+ REMOTE and CARE75+2 study arms will run concurrently. Participants who start on one study pathway will remain on that pathway. | |

Continued

| Table 1 | Continued | | |
|---|---|---|---|
| | **CARE75+** | **CARE75+REMOTE** | **CARE75+2** |

CARE75+, Community Ageing Research 75+; GP, General Practice; NHS, National Health Service; REC, Research Ethics Committee.

## DESIGN
A UK multisite, community-based cohort study designed using Trial within Cohorts (TWiCs) methods.[21]

## Inclusion criteria
Community-dwelling older people aged≥75 years.

## Exclusion criteria
People with an established life-limiting condition, life expectancy of 3 months or less and people in receipt of end of life palliative care services will be excluded. Care home and nursing residents and people living at home who are bedbound will be excluded. However, we will attempt to follow-up people who transition to a care home during the course of the study if feasible to do so.

## Recruitment
We will work with general practices to identify and recruit participants from primary care patient lists. Following initial piloting of CARE75+REMOTE with participants in Bradford, we will extend recruitment to other practices across England, using the skills and experience of staff within the NIHR Clinical Research Networks (CRN). Proposed CRN sites include: Thames Valley & South Midlands, North East and North Cumbria, South West Peninsula, West Midlands and Yorkshire & Humber. These regions provide geographic and socioeconomic diversity and contrasting urban/rural communities. See figure 1 for The CARE75+REMOTE recruitment flow and assessment schedule.

## Initial participant contact
Potential participants will be posted a study invitation pack containing a letter of invitation from the study team, a letter from their general practice and a brief user-friendly participant information leaflet to outline the study. The concise information sheet will include the

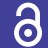

**Figure 1** The Community Ageing Research 75+(CARE75+) REMOTE study recruitment flow and assessment schedule.

names of the CARE75+REMOTE study lead and study manager, and the local researchers who will make the first telephone contact with the potential participant. Those not interested in participating in the study will be advised to contact their general practice to opt out of further contact, thus replicating the successful recruitment methodology adopted in the first iteration of CARE75+.[9]

If potential participants do not opt out of being contacted, their contact details will be provided to the research team approximately 2 weeks later via a secure email system. These details will be uploaded onto a bespoke CARE75+data application developed by Tiger-Team (www.tigerteam.co.uk). The researcher named on the brief information sheet will follow up with a telephone call to enquire whether the information has been received and to discuss the study in more detail. If interest in participation is expressed, the researcher will post out comprehensive study information and arrange a time to phone the participant back. During the follow-up call, verbal informed consent will be sought if the potential participant wishes to proceed.

### Monitoring numbers that opt out/do not consent
We will ask general practices to record the number of people who opt out of their contact details being forwarded to the research team. We will also monitor the number of people who do not opt out but: cannot be contacted by telephone; do not want to proceed (and the reasons if reported) or are unable to proceed (eg, due to severe hearing impairment or lacking capacity and are unable to identify a consultee to discuss participation).

### Participant consent
A paper consent form will be posted to participants for completion and return to the study team. Alternatively, if the person is unable to get to the post box, the researcher will provide the option of consent by telephone, which will be audio recorded using encrypted devices with audio files stored securely. A transcript of the recording will be sent to the participant if requested. Participant consent will include twelve 12 mandatory items and 6 optional items (table 2).

### Assessments
The CARE75+REMOTE model includes demographic information and validated instruments for assessments of frailty, mood, health-related quality of life, independence, comorbidities, medications as well as health and social care use and support. CARE75+REMOTE includes measures with the necessary validity, reliability and responsiveness for applied epidemiological investigation and as validated outcome measures for future embedded randomised controlled trials (RCT) of interventions to improve outcomes for those in later life.

### CARE75+ REMOTE assessments
► Demographic data including personal and family information.

► Instrumental activities of daily living (Nottingham Extended Activities of Daily Living, NEADL).[22] The NEADL includes questions on everyday activities in the domains of mobility, kitchen, domestic and leisure and is scored between 0 and 66, with higher scores indicating greater independence.

► Health-related quality of life, measured using the EuroQol Five Dimension Health Questionnaire (five-level version) EQ-5D-5L.[23] The EQ-5D-5L dimensions are: mobility, self-care, usual activities, pain/discomfort and anxiety/depression. Each dimension has five levels of severity: no problems, slight problems, moderate problems, severe problems and extreme problems. The scores for each of the five dimensions are combined in a five-digit number, representing health status that can be converted into a utility index (0 for dead, 1 for perfect health and negative values for states worse than death).

► Short-Form-12 item Health Survey, V.2 (SF12v2).[24] The 12 SF12v2 items are used to measure eight domains of health-related quality of life. The domains include physical functioning, role-physical; bodily pain; general health; vitality; social functioning; role emotional and mental health. The information obtained from the eight health domain scales is aggregated to generate physical component summary and mental component summary scores as overall measures of physical and mental health-related quality of life.

► Depression, measured using the Personal Health Questionnaire Depression Scale (PHQ-8).[25] The PHQ-8 case finding instrument assesses the severity of depression by asking eight questions to ascertain mood over the previous 2 weeks. There are four response options and the score is the sum of the eight items. A score of 10 or greater is considered major depression, 20 or more is severe major depression.

► Anxiety measured using the Generalised Anxiety Disorder 2-item (GAD-2), a two-item case finding tool for generalised anxiety disorder, with four response options. The GAD-2 score is obtained by summing the scores from each question.[26] A score of three points is the cut point for identifying possible cases of anxiety.

► Cognition measured using the Mini Montreal Cognitive Assessment V.2.1, and administered by telephone.[27]

► Health and social care use
  – Community-based health and social services (in the last 3 months)
  – Usual method of transport to services
  – Community-based social care services, including respite care (in the last 3 months)
  – Hospital services (in the last 3 months) including accident and emergency; acute hospital admissions; hospital outpatients (and transport to these services).

► Aids and adaptations
  – Type of aid, who provided it and the cost

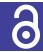

| Table 2 | Mandatory and optional consent items for CARE75+REMOTE | |
|---|---|---|
| **Mandatory consent items** | **Optional consent items** |
| (1) Have read the participant information sheet and been given a full verbal explanation of the study. | (1) A member of the study team to review records that may be held by social services about use of their services. |
| (2) Have been given enough time to think about the information. | (2) Researchers asking if I am interested in taking part in future studies that have been approved by the CARE75+oversight group and providing them with my contact details (name, address, phone number) and very limited information about me that will help them determine my suitability for future studies. |
| (3) Understand the reason for the research and what will happen if I take part. | (3) The researchers accessing and using data about me, as described above, in the event of my death. |
| (4) Understand that I am free to withdraw from the research at any time. | (4) The researchers informing my GP of my participation in the study. |
| (5) Understand that a member of the study team will review my medical records (primary care and hospital records), including my electronic health record. | (5) The researchers informing my GP if I am assessed as having an unmet health need. |
| (6) Understand that sections of my medical notes and data, including electronic health records, may be looked at by responsible individuals from the NHS Trust or regulatory authorities where it is relevant to my taking part in this research. I give permission for these individuals to access my records. | (6) If my capacity to make informed decisions should deteriorate during the course of this study I wish to remain a participant. |
| (7) Understand that all information about me will be kept strictly confidential and that information will be anonymous except in cases of safeguarding.<br><br>(8) Agree to my anonymous data being analysed by members of the research team and also by researcher teams outside of the team (this could include researchers from other universities) that have been approved by the CARE75+data review committee.<br><br>(9) Agree to the researchers conducting over the phone assessments with me at baseline (my first assessment) and then for me to undertake an assessment at 6, 12, 18, 24, 30 and 36 months (options of telephone, postal questionnaire, or web submission).<br><br>(10) Understand that even if I withdraw from the above study, the data already collected about me will be used to analyse the results of the study, unless I specifically withdraw consent for this. I understand that my identity will remain anonymous.<br><br>(11) Agree to my details, including my name and address, and a copy of this consent form to be stored at Bradford Institute for Health Research at Bradford Royal Infirmary for the sole purpose of managing this study.<br><br>(12) Agree for my personal details (eg, NHS number, date of birth, postcode, sex and initials) to be used to link study data with current and future data sources. These include (but are not limited to) primary and secondary care electronic health record systems (eg, SystmOne, EMISWeb, Vision, Cerner EPR), health (eg, Hospital Episode Statistics, Biologics Register, Cancer Registry), social services (eg, Community Care Statistics), environmental (eg, Air Pollution), economic (eg, Work and Pensions Longitudinal Study, Indices of Multiple Deprivation) and research databases (eg, UK Biobank, ResearchOne, CPRD, Q Research, ELSA) (data linkage). | |

GP - General Practice
NHS - National Health Service
EPR - Electronic Patient Record
ELSA - English Longitudinal Study of Ageing
CPRD - Clinical Practice Research Datalink

CARE75+, Community Ageing Research 75+.

– Type of adaptation, who provided it and the cost.
► Informal support (provided by friends and family) and the approximate time (hours/min).
► Prescribed medication information (name, dose, frequency). This will be collected from the primary care electronic health record (EHR) or via data linkage.

► Frailty will be assessed via the electronic Frailty Index (eFI) score, obtained from primary care EHR data.[28] The eFI is based on the cumulative deficit model of frailty, including 36 variables recorded in the primary-care EHR as part of routine care. The eFI score is calculated as an equally weighted proportion of the number of deficits present in an individual relative

to the total possible. The eFI enables identification of frailty categories (fit, mild frailty, moderate frailty, severe frailty). A cumulative deficit frailty index will also be operationalised using variables collected as part of the assessment schedule.

► Information on comorbidities will be collected via the primary care EHR or via data linkage.

### Assessment schedule

Participants will be assessed at baseline with follow-up assessments at 6, 12, 18, 24, 30 and 36 months. The baseline assessment will be conducted by telephone to capture demographic and family data and familiarise participants with the assessment questions. Follow-up assessments will be conducted by telephone, web submission or by a postal questionnaire, depending on participant preference. A phone call will be made a couple of weeks prior to each assessment time point to ensure that the person is happy to proceed and identify their chosen method of assessment.

The assessment schedule for CARE75+ has been carefully designed to accelerate the frailty translational research pathway by aligning epidemiological investigation with the typical follow-up schedule for feasibility and definitive trials of interventions.

### Face-to-face follow-up for some participants

We anticipate that some participants may have health or cognitive deterioration over their time in the study to an extent that they may struggle with the remote methods (telephone, postal or web submission) of assessments. For those participants, we will conduct follow-up assessments in person, in their homes, if feasible and safe to do so.

### Sample size

As of 2020, 1323 participants have consented to CARE75+. The aim is to recruit a further 1200 participants to achieve an approximate sample size of 2500 by September 2024. This will include participants recruited for CARE75+ REMOTE and for the planned full assessment schedule for CARE75+2. The latter will be implemented at some sites when face-face-face assessments can be resumed and sites have the necessary capacity.

This proposed sample size will provide a comprehensive dataset for applied epidemiological research, health and socioeconomic modelling and a recruitment platform for additional studies (substudies), including qualitative studies as well as RCTs using TwiCs methods.[21] Therefore, the recruitment target is based on appropriate sample size calculations for pilot RCTs of interventions to inform the design of future definitive RCTs alongside applied epidemiological investigation of modifiable components of frailty. Previous observational studies involving older people with frailty have identified that between 600 and 1000 participants are required for reliable estimates of the main effects.[29]

### Plans to promote participant retention and complete follow-up

We will seek broad and enduring consent for data linkage and use of collected data following withdrawal or death, aligned with Medical Research Council guidelines for maximising the use of cohort data.[30] Newsletters will be posted to participants at least two times a year to provide study updates and encourage continued engagement. We will hold annual engagement events, where feasible to do so, and promote the study locally via site affiliated and local forums.

### Data entry, coding, security and storage

The electronic data capture application is provided by TigerTeam (www.tigerteam.co.uk). It will comprise two main components: a data collection application (DCA) and back office system (BOS) containing personal identifiable information. The application will run on the Microsoft Windows platform using an encrypted embedded database to temporarily store data. The BOS database will be on a Microsoft SQL server hosted at Bradford Teaching Hospitals NHS Foundation Trust (BTHFT). Named researchers will have access to the individual details only while data collection takes place. A participant's details will only be released to one researcher at a time via the BOS management system. Access to modules and functions of both the DCA and BOS will be governed by usernames, passwords and role-specific access permissions, to maximise data security.

Remote site data (outside BTHFT) and the web-submission forms will be transferred to the BIHR-CARE75+ database via the web application auecr.bradfordhospitals.nhs.uk hosted on the web server bhts-bihrweb. The site will be protected by Secure Sockets Layer (SSL) certificates, to encrypt the transfer of data over the internet. Access to the web application on the server will be restricted and protected by the Threat Management Gateway software and SSL certificates. Remote site administrators and researchers will only have access to their local participants.

The chief investigator, project manager, database manager and core team members from CARE75+ based at BTHFT with Super Administrator roles will have access to all data, at all levels for administration and governance purposes. Local teams will be limited to access data of their participants only.

Researchers will have a maximum of three participants available on portable devices (laptops) at any one time. Statisticians and other members of the CARE75+ research team will only have access to pseudo anonymised that is, those with unique identifiers for use in data linkage or anonymous data. Individual participants will be limited to access to a blank follow-up questionnaire to complete and submit if undertaking web submission.

### Data quality

Data quality will be enhanced by integral features of the data capture software, which will identify missing data and outlying values in real time. The software will automatically

calculate the total scores for composite assessments. This will increase research efficiency and research data quality by reducing resource required for data cleansing, coding for analysis and reduce input errors.

## Statistical methods

We will use a range of appropriate statistical methods to investigate a broad range of research questions focusing on frailty, coexisting physical and mental health problems and independence in later life, including descriptive epidemiology, prognostic modelling methods, causal epidemiology, transition and trajectory modelling. We will investigate health and social care resource use for people with frailty, multimorbidity and disability using economic modelling methods. Submission of detailed statistical analysis plans will be required as part of the data application process for researchers to obtain access to CARE75+REMOTE data.

## Missing data

Methods for dealing with missing data will depend on the amount of missing data and patterns of missingness for individual variables as part of individual analyses. We will undertake sensitivity analysis to investigate the impact of missing data and we will explore the use of appropriate imputation methods.

## Ethics and dissemination

CARE75+REMOTE is an observational study with low risk to participants. Cohort governance will be provided by the NIHR ARCYH (https://www.arc-yh.nihr.ac.uk/what-we-do/older-people) Operational Group comprising of theme leads, theme manager, project managers and coapplicants. The original CARE75+protocol was approved by the National Research Ethics Service (NRES) Committee Yorkshire and the Humber—Bradford Leeds, October 2014 (14/YH/1120). The CARE75+REMOTE protocol (amendment number 13) was approved on 18 November 2020.

## Data sharing and data access

The protocol and participant-level dataset will be made available to not-for-profit investigators. Enquiries will be made to the CARE75+chief investigator or CARE75+manager. Data request forms are available at https://www.bradfordresearch.nhs.uk/care75/data-request/. Requests will be reviewed by the CARE75+data review committee, which comprises the study lead, study manager, academics within the Academic Unit for Ageing and Stroke Research, a PPI lay representative and a statistician. For studies that request contact details for access to participants, they will first be invited to present their proposal to the established FOG.[19] This is to ensure that the study is an appropriate use of the CARE75+resource, being mindful that too many requests may be detrimental to the study overall if participants have multiple requests for their time or requests to participate in studies not relevant to people in later life.

Bradford Teaching Hospitals NHS Foundation Trust is the data controller for CARE75+. Data will be made available to external researchers in accordance with CARE75+data sharing protocols following review, completion of a data request form and completion of a data sharing agreement between BTHFT and the recipient university or organisation.

## Dissemination policy

Study results will be disseminated in peer-reviewed scientific journals and submitted to local, national and international scientific conferences. Key results will be summarised and disseminated to study participants via newsletters, local older people's publication (eg, Voice magazine, Age UK) and local engagement events. Links to publications will be shared on the CARE75+website (https://www.bradfordresearch.nhs.uk/care75/) and the ARC Yorkshire and Humber website (https://www.arc-yh.nihr.ac.uk/home). Research outputs using CARE75+study data or including CARE75+participants will be required to acknowledge the data source and funder.

We will use the cross-ARC network and additional implementation expertise within ARC for https://www.arc-yh.nihr.ac.uk/ to maximise impact of research outputs. We will share results with an established Research Implementation Advisory Group that runs alongside the frailty research programme. The group has broad membership, with regional representation across NHS commissioner and provider organisations, adult social care, public health, voluntary sector and national policy representation. This implementation advisory group will act as a 'pull' to facilitate getting frailty research findings into routine practice.

## Ancillary and poststudy care

Some participants may be identified with unmet care needs and may wish to discuss these with the researcher. Where applicable, researchers will signpost participants to local statutory and voluntary organisations (eg, Age UK), or request a General Practice (GP) referral for social services assessment, so that appropriate plans can be made for ongoing care. Safeguarding issues identified during the assessment will be reported to the research project manager who will then take advice from the Adult Safeguarding Coordinator in the relevant local authorities.

## DISCUSSION

CARE75+REMOTE is designed using a remote model that does not require face-to-face interactions to support data collection in the extreme circumstances of a global pandemic, but to also provide a highly efficient, sustainable model of data collection. Remote data collection is less costly than face-to-face, an important consideration for publicly funded research, particularly in uncertain economic times. In line with the original CARE75+study,

CARE75+REMOTE is designed using TwiCs methods[21] to align applied epidemiological research with clinical trials of interventions, potentially accelerating the translational research pathway in the important area of improving health for older people living with multimorbidity and frailty. We plan to include a concise range of measurements to capture important outcomes on mood, independence, quality of life, health and social care use and frailty, including through use of the primary care EHR and routine data linkage.[28]

The new remote model, CARE75+REMOTE, is not without challenges and we are mindful that a lack of face-to-face interactions may be less attractive to some participants than the original CARE75+model. However, we have already used a remote model to collect data for CARE75+follow ups, through web submission or telephone, and this has proven to be an effective form of data collection. We will monitor recruitment rates to ensure that we are still able to successfully recruit older people across the frailty spectrum, particularly those with advanced frailty, and compare the characteristics of participants recruited using remote, compared with face-to-face methods.

For the CARE75+REMOTE pathway, we have developed a protocol that includes key outcomes and is not overly burdensome for self-completion or completion by telephone. However, we are aware that there are other important outcomes that would be pertinent to include. Therefore, we will monitor takeup and completion rates, alongside researcher feedback and consider the inclusion of further outcome measures. These could include, for example, loneliness or participation, or other measures considered a priority by academic partners and lay members.

The original CARE75+methodology was previously very successful in recruiting older people from the South Asian population[31] as a historically underserved population.[15] We will continue to include researchers with the appropriate community language skills to support recruitment of South Asian participants and will monitor take-up to see whether the previous success is maintained with the remote recruitment and assessment protocol. We aim to include people from different ethnocultural backgrounds where feasible to do so, mindful of language constraints.

We will provide regular participant newsletters, share results and host local events to facilitate continued engagement over the course of the study. We envisage this to be particularly important for a cohort that will otherwise lack face-to-face contact with researchers.

The 2020 COVID-19 pandemic has had a substantial impact on non-COVID-19 research and has particularly impacted research, which requires face-to-face assessment, and research involving older people.[20] Many older people are considered vulnerable to COVID-19 due to their age and due to health conditions prevalent within their age group. As such, face-to-face data collection techniques are likely to be affected by the pandemic for some time due to a combination of government imposed restrictions, due to COVID-19, and older people themselves limiting unnecessary in-person interactions due to concerns about COVID-19. The COVID-19 pandemic has highlighted the need for timely, accessible information on the health and well-being of older populations. The remote model will support rapid, responsive data collection. CARE75+REMOTE will contribute to a cohort with high strategic relevance, which will help shape UK and international research policy on ageing and frailty.

**Author affiliations**
[1]Academic Unit for Ageing and Stroke Research, Bradford Institute for Health Research, Bradford, UK
[2]Centre for Applied Dementia Studies, Faculty of Health Studies, University of Bradford, Bradford, UK
[3]College of Medicine and Health, University of Exeter, Exeter, UK
[4]School of Medicine, University of Keele, Keele, UK
[5]Academic Unit for Ageing and Stroke Research, University of Leeds, Faculty of Medicine and Health, Leeds, UK
[6]School of Healthcare, Faculty of Medicine and Health, University of Leeds, Leeds, UK
[7]School of Health and Related Research (ScHARR), The University of Sheffield, Sheffield, UK
[8]Population Health Sciences Institute, Newcastle University, Newcastle upon Tyne, UK

**Contributors** LB: substantial contribution to the conception and design of the work; drafting the work and critical revisions; approval of final manuscript; accountable for all aspects of the work. AH: substantial contribution to the conception and design of the work; drafting the work and critical revisions; approval of final manuscript; accountable for all aspects of the work. CQ: substantial contribution to the conception and design of the work; drafting the work and critical revisions; approval of final manuscript; accountable for all aspects of the work. VG: substantial contribution to the conception and design of the work; drafting the work and critical revisions; approval of final manuscript; accountable for all aspects of the work. CC-G: substantial contribution to the conception and design of the work; drafting the work and critical revisions; approval of final manuscript; accountable for all aspects of the work. IJ: substantial contribution to the conception and design of the work; drafting the work and critical revisions; approval of final manuscript; accountable for all aspects of the work. FM: substantial contribution to the conception and design of the work; drafting the work and critical revisions; approval of final manuscript; accountable for all aspects of the work. SH: substantial contribution to the conception and design of the work; drafting the work and critical revisions; approval of final manuscript; accountable for all aspects of the work. CB: substantial contribution to the conception and design of the work; drafting the work and critical revisions; approval of final manuscript; accountable for all aspects of the work. VG: substantial contribution to the conception and design of the work; drafting the work and critical revisions; approval of final manuscript; accountable for all aspects of the work. KB: substantial contribution to the conception and design of the work; drafting the work and critical revisions; approval of final manuscript; accountable for all aspects of the work. AD-O: substantial contribution to the conception and design of the work; drafting the work and critical revisions; approval of final manuscript; accountable for all aspects of the work. KS: substantial contribution to the conception and design of the work; drafting the work and critical revisions; approval of final manuscript; accountable for all aspects of the work. BH: substantial contribution to the conception and design of the work; drafting the work and critical revisions; approval of final manuscript; accountable for all aspects of the work. TAY: substantial contribution to the conception and design of the work; drafting the work and critical revisions; approval of final manuscript; accountable for all aspects of the work. RH: substantial contribution to the conception and design of the work; drafting the work and critical revisions; approval of final manuscript; accountable for all aspects of the work. ET: substantial contribution to the conception and design of the work; drafting the work and critical revisions; approval of final manuscript; accountable for all aspects of the work. AC: substantial contribution to

the conception and design of the work; acquisition of data for the work; drafting the work and critical revisions; approval of final manuscript; accountable for all aspects of the work.

**Funding** This research was funded by the NIHR Applied Research Collaboration (ARC) Yorkshire and Humber-https://www.arc-yh.nihr.ac.uk/ (study funding number NIHR200166), also supported by NIHR ARC South West Peninsula, ARC West Midlands and ARC North East and North Cumbria.

**Disclaimer** The views expressed in this publication are those of the author(s) and not necessarily those of the NHS, the National Institute for Health Research or the Department of Health and Social Care.

**Competing interests** None declared.

**Patient and public involvement** Patients and/or the public were involved in the design, or conduct, or reporting, or dissemination plans of this research. Refer to the Methods section for further details.

**Patient consent for publication** Not applicable.

**Provenance and peer review** Not commissioned; externally peer reviewed.

**ORCID iDs**
Lesley Brown http://orcid.org/0000-0001-5499-9145
Victoria Goodwin http://orcid.org/0000-0003-3860-9607
Karen Spilsbury http://orcid.org/0000-0002-6908-0032
Barbara Hanratty http://orcid.org/0000-0002-3122-7190
Andrew Clegg http://orcid.org/0000-0001-5972-1097

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
