## [Reviewer comments · BMJ Open]

ARTICLE DETAILS

TITLE (PROVISIONAL)	The Community Ageing Research 75+ (CARE75+) REMOTE Study: a remote model of recruitment and assessment of the health, wellbeing and social circumstances of older people
AUTHORS	Brown, Lesley; Heaven, Anne; Quinn, Catherine; Goodwin, Victoria; Chew-Graham, Carolyn; Mahmood, Farhat; Hallas, Sarah; Jacob, Ikhlaq; Brundle, Caroline; Best, Kate; Daffu-O'Reilly, Amrit; Spilsbury, Karen; Young, Tracey; Hawkins, Rebecca; Hanratty, Barbara; Teale, Elizabeth; Clegg, Andrew

VERSION 1 – REVIEW

REVIEWER	Sanchez-Garcia, Sergio National Medical Center Mexican Social Security Institute Mexico City, Unidad de Investigación Epidemiológica y Servicios de Salud, Área envejecimiento
REVIEW RETURNED	16-Apr-2021

GENERAL COMMENTS	This manuscript aims to report the Community Ageing Research 75+ (CARE75+) Study is a longitudinal cohort study that collects extensive health and social data, with a focus on frailty, independence and quality of life in older age. CARE75+ was the first international experimental frailty research cohort designed using Trial within Cohorts (TwiCs) methodology, aligning epidemiological research with clinical trial evaluation of interventions to improve the health and well-being of older people. CARE75+ REMOTE is an extension of CARE75+ that uses a remote model that does not require face-to-face interactions for data collection in the current circumstances of a global pandemic, and will provide an efficient and sustainable data collection model. Unfortunately, in the form in which it is presented it does not allow the reader to learn about the Community Ageing Research 75+ (CARE75+) Study. Therefore, it is proposed to include some tables or figures that will allow the reader to get to know the protocol in a pleasant way. I consider that they are in the possibility of presenting the characteristic of the participants. It is recommended to consult the following article Ruiz-Arregui L, Ávila-Funes JA, Amieva H, Borges-Yáñez SA, et al. The Coyoacán Cohort Study: design, methodology, and participant characteristics of a Mexican study on nutritional and psychosocial markers of frailty. J Frailty Aging. 2013;2(2):68-76. doi: 10.14283/jfa.2013.11.
---

REVIEWER	Stanaway, Fiona University of Sydney, School of Public Health
REVIEW RETURNED	18-May-2021

GENERAL COMMENTS	This a well written protocol and I only have a few concerns/comments to be addressed: Strengths and limitations section  • Point two about when assessments will be conducted and what these will include is not a clear strength. I would suggest removing this. An alternative strength could be to highlight that outcome measures were agreed following discussion with academic partners and lay representatives as mentioned on page 5 • Including participants from a variety of ethnic backgrounds is a clear strength but I am unclear from the protocol how this will be achieved e.g. use of translated materials/interpreters– other than the evidence of engagement and success with recruiting members of South Asian communities. If those of South Asian backgrounds are likely to be the main ethnic minority persons recruited then it might be better to modify this statement to better reflect this. Introduction  • I found it took quite some time to work out the relationship between the original CARE75+ study and the new CARE75+ REMOTE. There could be better clarity in this section that the new protocol relates to the new methods for recruitment of new participants as well as the follow-up of participants that have already been recruited into CARE75+. CARE75+ REMOTE assessments  • Whilst I am hesitant to criticise the listed assessments given that the outcomes to be measured were selected based on discussion with both academic partners and lay representatives, a key lack on terms of outcomes is the assessment of social engagement, participation and loneliness. As well as being important outcomes in older adults and a key part of healthy ageing, they have also been greatly impacted by COVID-19. They are also likely to be important drivers of the depression and anxiety symptoms being measured. Is there a reason why no such measures have not been considered as outcomes? • Are there differences in outcome measures in CARE75+ REMOTE compared to CARE75+?
--

VERSION 1 – AUTHOR RESPONSE

Response to reviewer 1

We have given your comments some consideration and have now included an additional table (Table 1, page 6) which provides information about the different CARE75+ study pathways in a format we hope results in an easier read.

We decided not to include baseline characteristics as we are specifically describing the protocol for the remote study arm and we do not want to conflate results from the original study methodology with a detailed description of the remote study pathway. Following earlier correspondence with the BMJ journal editor, we intend to submit a cohort profile paper which is structured to allow a detailed description of the cohort at baseline and follow-up.

Response to reviewer 2

Strengths and limitations

1. Thank you for your suggestion to remove the item about assessment time-points. We have replaced the item with your proposed suggestion: Outcome measures were agreed following discussion with academic partners and lay representatives. This is reported in the Strengths and Limitations section on page 4.

2. Thank you for your comment regarding the inclusion of participants from different ethnic backgrounds. We have successfully recruited from the South Asian population as we have researchers who speak a number of South Asian community languages; written material is less relevant to this population as most of the older generation do not read or write and some community languages do not have a written format. On page 18 in the discussion we report the use of researchers with the appropriate language skills for the south Asian population. However, in view of your suggestion, we have modified the study strength to be more specific: CARE75+ REMOTE will continue to recruit underserved populations including participants from the South Asian community and those with advanced frailty. This is reported in the Strengths and Limitations section on page 4.

Additionally, we have added the following statement to the discussion: We aim to include people from different ethno-cultural backgrounds where feasible to do so, mindful of language constraints. This has been added to page 18.

Introduction

3. Thank you for your point regarding the unclear relationship between the original CARE75+ study pathway and the new CARE75+ REMOTE pathway. We have now included a table which details the study pathways. We hope this provides clarity. Table 1, page 6.

4. CARE75+ REMOTE assessments and lack of inclusion of assessment of social engagement, participation and loneliness.

I think you raise a very good point and we agree that social engagement, participation and loneliness are very important outcomes. When we resume a full CARE75+ protocol, with the next iteration - CARE75+2 - we will include the outcome of loneliness. This is reported in the outcome domain section of Table 1, page 6.

In terms of CARE75+ REMOTE, we deliberately kept the protocol very tight, mindful of the potential for burden when asking questions by telephone (which seems to be the most popular option for participants to date). However, initial feedback from the researchers suggest that the REMOTE protocol is working well and does not appear overly burdensome. As such, we will monitor for a few more months and then consider introducing some additional measures. If so, loneliness will be included.

In light of your suggestion, I have added the following paragraph to the discussion on page on page 18.

For the CARE75+ REMOTE pathway we developed a protocol which included key outcomes and was not overly burdensome for self-completion or completion by telephone. However, we are aware that there are other important outcomes that would be pertinent to include. Therefore, we will monitor take-up and completion rates, alongside researcher feedback and consider the inclusion of further outcomes measures. These could include, for example, loneliness or participation, or other measures considered a priority by academic partners and lay members.

5. Are there differences in outcome measures in CARE75+ REMOTE compared to CARE75+? The addition of Table1, page 6 lists the outcomes collected across all study arms, and those that are specific to the pathway.

Additional changes not proposed by the reviewers

Since the original draft, we have included a short telephone assessment of cognition using the Mini Montreal Cognitive Assessment (Mini MoCA) Version 2.1. This is reported on page 12.

We anticipate working with additional Clinical Research Network (CRN) sites to those original reported. Therefore, we have added the West Midlands and the Yorkshire and Humber to our list of sites. Page 8.

Also added NIHR ARC South West Peninsula in the acknowledgments.

VERSION 2 – REVIEW

REVIEWER	Stanaway, Fiona University of Sydney, School of Public Health
REVIEW RETURNED	13-Sep-2021

GENERAL COMMENTS	The authors have responded well to all reviewers' comments and I have no further concerns.
--